# Anti-nucleocapsid and anti-spike antibody trajectories in people with post-covid condition versus acute-only infections: a nested longitudinal case-control study within the Virus Watch prospective cohort

People with Post-Covid Condition (PCC) may demonstrate aberrant immune responses post-infection; however, serological follow-up studies are limited. We aim to compare SARS-CoV-2 serological responses to infection and vaccination in people who develop PCC versus those with an acute infection only. Participants (*n* = 2010) are a sub-cohort of the Virus Watch community cohort in England who provided monthly finger-prick serological samples. We compare the likelihood of post-infection seroconversion using logistic mixed models and the trajectories of anti-nucleocapsid (anti-N) and anti-spike (anti-S) antibodies using linear mixed models. Participants who developed PCC (*n* = 394) have 1.8x the odds of post-infection seroconversion for anti-N antibodies compared to those with an acute infection only (*n* = 1616) (odds ratio= 1.81 (95% confidence interval (CI) 1.16-2.90); however, these results are moderated by vaccination status and variant – with differences observed in pre-Omicron, unvaccinated participants. Anti-N levels, however, were elevated within 200 days post-infection in people with PCC compared to those without, after accounting for variant and vaccination status. Vaccination response (anti-S) pre- or post-infection did not systematically differ between groups. People with PCC demonstrate persistently higher anti-N antibody levels following primary infection compared to those with an acute infection only. These findings extend emerging evidence around infection-related immune activation and PCC.

While vaccination and effective treatments have reduced the threat from acute infection with Severe Acute Respiratory Syndrome Coronavirus 2 (SARS-CoV-2), ~10-30% of people with mild-moderate infections and up to 50% of people with severe infections are left with chronic, interfering post-infectious symptoms[1,2]. Post-Covid Condition (PCC) or Long Covid is a heterogeneous condition that can affect a range of organ systems, with commonly-reported symptoms including fatigue, dyspnoea, myalgia and/or arthralgia, and cognitive dysfunction[3]. Understanding of the mechanisms underlying the development and maintenance of PCC is limited but evolving. Current evidence suggests multiple mutually non-exclusive mechanisms, including SARS-CoV-2 persistence and/or reactivation of other latent viruses, inflammatory pathologies and autoimmunity, and coagulopathies and endothelial dysfunction[1,4]. The specific underlying

✉e-mail: sarah.beale@ucl.ac.uk

mechanism(s) may vary by case and over time and interact with one another to produce the diverse range of symptoms reported[1,4]. Aberrant immune response to SARS-CoV-2 may underlie the transition from acute infection to chronic illness, and longitudinal studies investigating this transition are required. Additionally, identifying immune correlates of PCC may provide useful clinical markers to assist with case identification as well as understanding aetiology.

While some individuals clear SARS-CoV-2 through their initial cellular and humoral innate immune response, the majority of people develop an adaptive immune response aimed at resolving the infection and leading to long-term immunological memory[5]. Antibodies produced as part of this adaptive response include long-term Immunoglobulin G (IgG) antibodies targeting the SARS-CoV-2 spike and nucleocapsid structural proteins[5]. The spike protein enables viral entry into host cells[6,7] and is the target of most licensed COVID-19 vaccines; consequently, anti-spike (anti-S) antibodies are produced in response to both natural infection and vaccination[8]. The nucleocapsid protein is critical for the viral RNA genome packaging[6,7] and anti-nucleocapsid (anti-N) antibodies are produced in response to natural infection[9]. The degree and persistence of individuals' serological response to a SARS-CoV-2 immune challenge are influenced by a range of factors including age, sex, infection severity, comorbidities and immune-modulatory medications[10].

Early evidence suggests differences in the antibody response to infection in people who develop PCC compared to those who recover fully from SARS-CoV-2. Two-year post-infection follow-up of 31 people with PCC and 31 acute-only controls found that markers of inflammation and anti-S and anti-N antibodies were elevated in people with PCC, but normalised to the levels of the acute-only group after 12-24 months[11]; participants were infected prior to the availability of COVID-19 vaccines, but later post-vaccination antibody responses appeared to be similar between groups. Similarly, a cohort of people with mild SARS-CoV-2 infections who were unvaccinated at the time of infection found that anti-S levels were higher at three months in women with PCC compared to those with acute infections only, and that levels of anti-N antibody were similar between both groups[12]. A further cohort of 51 community cases followed-up twice for 5-6 months post-infection found persistently elevated anti-S and anti-N antibody levels in people with greater post-infectious symptomatology[13]. The direction of findings in the literature was mixed, with a single-site hospital cohort of 107 patients finding persistently lower anti-N levels and similar levels of anti-S in people with PCC compared to those with an acute infection only; however, all participants in this cohort received antivirals likely limiting generalizability of findings[14]. A further small, single-clinic study (n = 61) monitoring anti-N levels over eight months post-infection found no difference between people with and without PCC[14,15].

Differences in post-infection antibody responses in people with PCC may be indicative of immune activation[11] or the persistence of SARS-CoV-2 virus or antigens during the transition from acute to chronic illness and in the maintenance of PCC symptoms, and warrants further investigation. Small sample sizes in the current literature, which is largely based on single-site cohorts, may introduce uncertainty into the comparison of people with PCC versus those with acute infections only. Furthermore, current studies are based exclusively on early-pandemic infections in unvaccinated people, and investigation including infections from later pandemic periods is warranted and would allow delineation of differences in post-vaccination and post-infection antibody responses in people with PCC.

In this work, we demonstrate differences in SARS-CoV-2 antibody responses following primary infection in people with PCC compared to those with an acute infection only. The primary aim of this study was to compare the anti-S and anti-N antibody dynamics related to primary SARS-CoV-2 infections in people who developed PCC and those who did not. Participants were drawn from Virus Watch, a large UK community cohort including an adult sub-cohort who conducted monthly

finger-prick serological sampling between February 2021 and March 2022. Research questions were:
1. Do odds of sero-conversion and anti-N antibody trajectories following infection differ between people who developed PCC and those with an acute infection only?
2. Do anti-S antibody trajectories differ following vaccination and/or infection between people who developed PCC and those with an acute infection only?

## Results

Table 1 reports participants' demographic characteristics overall and by PCC status. Of the 2010 included participants, 20% (n = 394) developed PCC according to the WHO consensus definition following SARS-CoV-2 infection, while 80% (n = 1616) experienced an acute infection only. The 2010 participants submitted 9466 included samples, with a median of 5 samples submitted per participant (interquartile range 3-6). Supplementary Fig. 4 reports participant selection into the study. The number of samples within each regression model are reported overall and by PCC status in Supplementary Table 1; the majority of samples (79%, n = 7512) occurred prior to infection and were included in pre-infection, vaccination-related anti-S models only. Consequently, differences between participants' vaccination status at the time of infection (Table 1) and the number of participants in post-vaccination, post-infection anti-S models (Supplementary Table 1) were due to lower sample availability post-infection. Table 1 reports on all participants, including those with only pre-infection samples available.

### Anti-Nucleocapsid Seropositivity by PCC Status

Across the full follow-up period, people with PCC were more likely to demonstrate detectable anti-N seropositivity after accounting for the effects of age, sex, and clinical vulnerability (OR = 1.81 (95% CI 1.16-2.90, $p = 0.01$) compared to people with an acute infection only (see Supplementary Table 2 for odds ratios). Corresponding predicted probabilities (PP) for overall seroconversion were 84.7% (95% CI 78.9%-89.2%) amongst people with PCC versus 75.4% (95% CI 71.0% - 79.3%) for people with an acute infection only.

When disaggregated by time since infection (see Fig. 1 for predicted probabilities, and Supplementary Table 2 for odds ratios), people with PCC were more likely to demonstrate detectable anti-N seropositivity than people with an acute infection beyond the effect of chance during the following time periods: 60-89 days (OR = 2.26, 95% CI 1.07-5.25, $p = 0.04$), 90-119 days (OR = 3.86, 95% CI 1.89-10.00, $p = 0.003$), 120-269 days (OR = 3.84, 95% CI 2.02-7.80, $p < 0.001$), and 270+ days (OR = 4.45, 95% CI 2.08-10.00, $p < 0.001$). This corresponds with a pattern of increasing predicted probabilities of seropositivity during the initial periods following infection followed by a gradual decline in seropositivity in the acute-only group (78.3%, 95% CI 71.7-83.6% seropositive at 30-59 days versus 50.7%, 95% CI 39.1-62.3% seropositive at 270+ days) compared to persistently high seropositivity maintained across this period in the PCC group (85.2%, 95% CI 73.1-92.4% seropositive at 30-59 days and 82.1%, 95% CI 71.1-89.5% seropositive at 270+ days) (Fig. 2).

**Investigation into effect modification by sex, comorbidity status, vaccination status, and variant.** We found no evidence of an interaction between PCC status and sex or between PCC status and comorbidities beyond the effects of chance (Supplementary Table 3), indicating a similar effect of PCC status according to sex and comorbidity status. The corresponding stratified predicted probabilities of seroconversion (Table 2) demonstrate that in both male and female participants and in participants with and without comorbidities, people with PCC had a trend towards greater probability of seroconversion than those without; however, confidence intervals for predicted probability estimates overlapped for stratified models. We found evidence of an interaction between PCC status and variant of infection

## Table 1 | Participant Characteristics

| | Overall Cohort N = 2010 | Acute Only N = 1616 | Post-Covid Condition N = 394 |
|---|---|---|---|
| Post-Covid Condition Status - n (%), | | | |
| Acute Only | 1616 (80%) | - | - |
| Post-Covid Condition | 394 (20%) | - | - |
| Age - median (IQR[a]) | 64 (57, 70) | 65 (58,70) | 61 (53,69) |
| Sex - n (%), | | | |
| Female | 1280 (59%) | 928 (57%) | 282 (72%) |
| Male | 888 (41%) | 688 (43%) | 112 (28%) |
| Comorbidities - n (%), | | | |
| No | 1208 (60%) | 1012 (63%) | 196 (50%) |
| Yes | 802 (40%) | 604 (37%) | 198 (50%) |
| Vaccination Status at Time of Infection[b] - n (%), | | | |
| Unvaccinated[c] | 159 (7.9%) | 60 (3.7%) | 99 (25%) |
| 1 Dose[c] | 31 (1.5%) | 24 (1.5%) | 7 (2%) |
| 2 Doses | 192 (9.6%) | 145 (9.0%) | 47 (12%) |
| 3 Doses | 1460 (73%) | 1253 (78%) | 207 (53%) |
| >3 Doses[c] | 168 (8.5%) | 134 (8%) | 34 (9%) |
| Dominant Variant at Time of Infection[d] | | | |
| Wild Type | 67 (3.3%) | 34 (2.1%) | 33 (8.4%) |
| Alpha | 109 (5.4%) | 47 (2.9%) | 62 (16%) |
| Delta | 197 (9.8%) | 142 (8.8%) | 55 (14%) |
| Omicron BA.1 | 295 (15%) | 234 (14%) | 61 (15%) |
| Omicron BA.2 | 1341 (67%) | 1158 (72%) | 183 (46%) |
| Overall Number of Submitted Samples - median (IQR[a]) | 5 (3,6) | 5 (4,6) | 4 (2,6) |
| Number of Submitted Samples Post-Infection - median (IQR[a]) | 2 (1,5) | 2 (1,4) | 2 (1,6) |

[a]IQR = interquartile range, [b] for number of samples by vaccination status pre-infection please see Supplementary Table 1, [c]only included in anti-N model as insufficient number of samples for stratified anti-S analyses; [d]variant periods based on UK Office for National Statistics estimates https://www.ons.gov.uk/peoplepopulationandcommunity/healthandsocialcare/conditionsanddiseases/articles/coronaviruscovid19infectionsurveytechnicalarticlecumulativeincidenceofthenumberofpeoplewhohavebeeninfectedwithcovid19byvariantandageengland/9february2023#cumulative-incidence-by-period-variant-and-age.

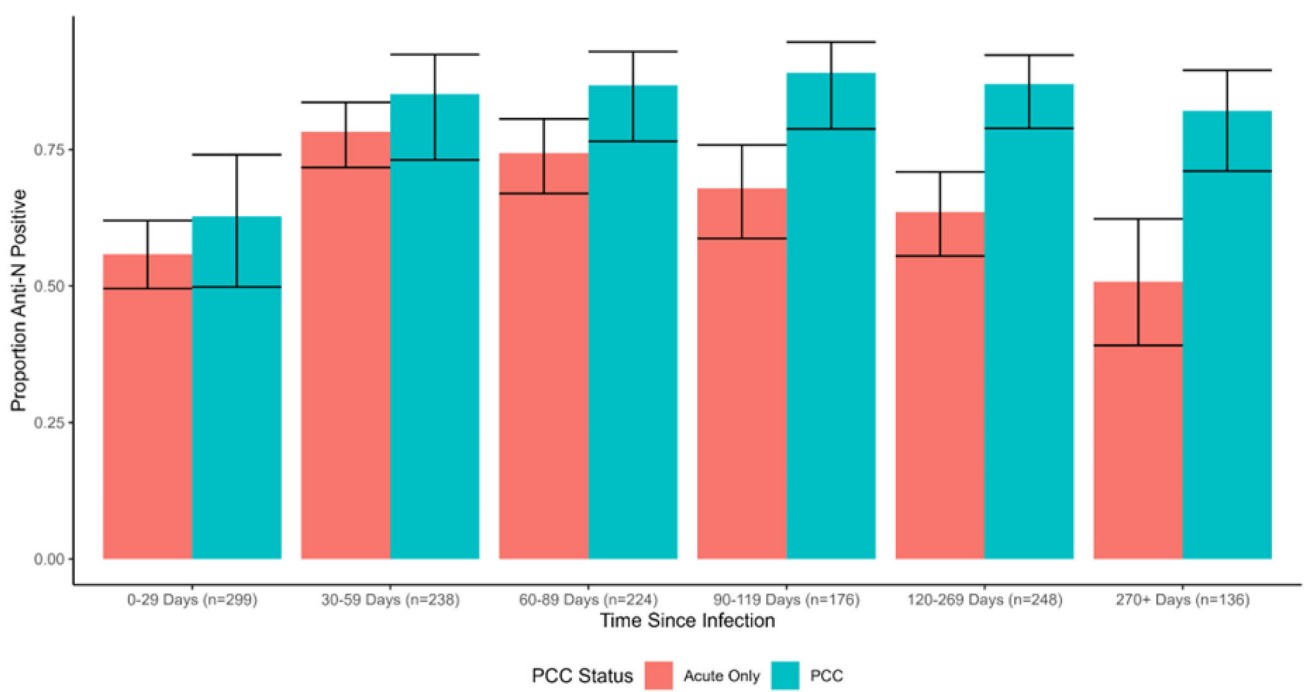

**Fig. 1 | Predicted Probabilities with 95% Confidence Intervals for Detectable Anti-N Seropositivity in People with PCC versus Acute Infection Only, by Time Since Infection. Note:** Please see Supplementary Table 2 for raw frequencies of seropositivity over time and for odds ratios from the binary logistic regression models.

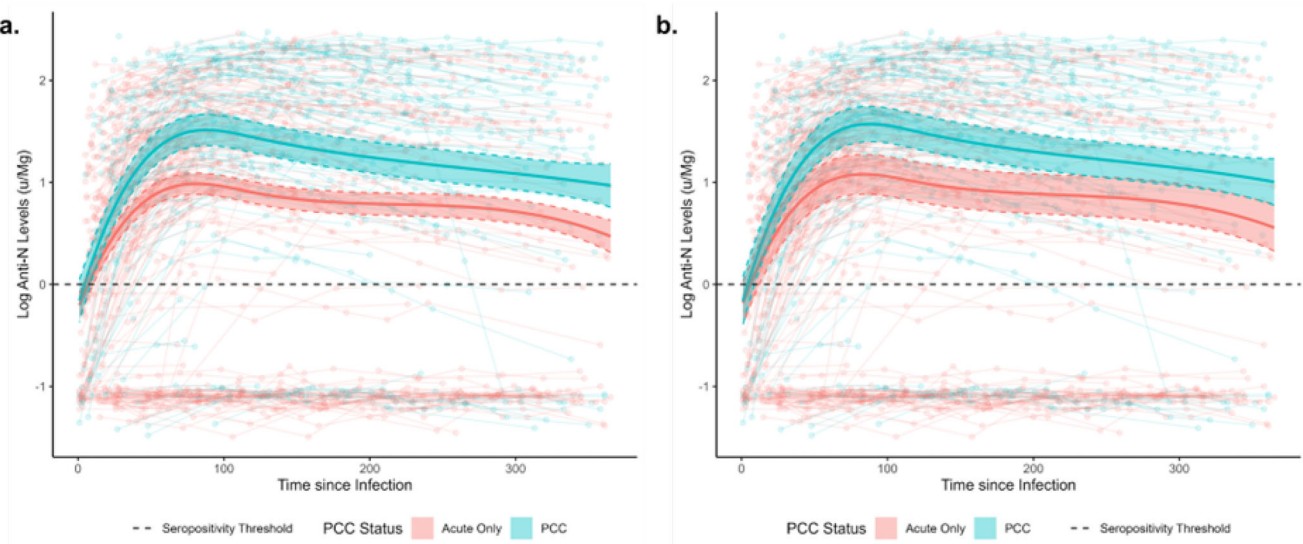

**Fig. 2 | Anti-Nucleocapsid Antibody Trajectories with 95% Confidence Intervals in Participants with Post-COVID Condition versus Acute Infection Only.** Panels present unadjusted (**a**) and adjusted estimates for variant and vaccination status (**b**).

**Note:** Anti-N= Anti-Nucleocapsid; PCC = Post-COVID Condition; solid lines indicate regression estimates for anti-N trajectories over time; scatterplot indicates raw anti-N trajectories for individual participants; *n* = 1094 samples from 604 participants.

(Supplementary Table 3), adjusted for vaccination status, with people with PCC demonstrating greater odds of seroconversion compared to those with an acute infection during the pre-Omicron period only (Table 2). Likewise, people with PCC were more likely to seroconvert than those with an acute infection only amongst the unvaccinated (Table 2), with no differences beyond the effects of chance amongst those who had received two or three vaccine doses (Table 2).

These investigations were included to investigate potential interactions between demographic and clinical variables and PCC status; for an appropriately powered and designed investigation of the main effects of sex on anti-N seropositivity, please see ref. 10.

### Log Anti-N Antibody Levels over Time by PCC Status

Anti-N antibody trajectories are illustrated in Fig. 2, with the solid lines illustrating regression estimates and the scatterplots illustrating raw trajectories. Individual regression coefficients are not interpretable in the presence of the interaction term and spline, and consequently results are presented graphically and quantified using estimated marginal predictions for key timepoints. Log antibody levels diverged by PCC status beyond the effect of chance from Day 24 post-infection, with an estimated log anti-N level of 0.73 (95% CI 0.57–0.88) in people who developed PCC versus 0.46 (95% CI 0.36–0.56) in those with an acute infection only. Estimated levels were higher in the PCC group for the remainder of the follow-up period, following a similar trajectory in both groups with an initial exponential rise followed by a gradual waning. Levels peaked on Day 88 post-infection in the PCC group (estimated log anti-N Level=1.51, 95% CI 1.36–1.66) and on Day 81 in the acute-only group (predicted log anti-N Level=0.99, 95% CI 0.88–1.09). By 365 days post-infection, estimated levels had waned to 0.97 (95% CI 0.76–1.18) in the PCC group versus 0.47 (0.31–0.62) in the acute-only group. When adjusted for vaccination status and variant of infection (Fig. 2b), similar trajectories and between-group differences were identified by PCC status, though differences were less pronounced and confidence intervals overlapped during later periods of follow-up beyond 200 days. Similar results were obtained in the sensitivity analysis limited to participants who seroconverted for anti-N (Supplementary Fig. 5).

**Investigation into Effect Modification by Sex and Comorbidity Status.** Figure 3 presents predicted anti-N trajectories stratified by sex and comorbidity status, based on regressions including three-way interaction terms and adjusted for vaccination status and variant of infection (see Statistical Methods). Likelihood ratio tests did not indicate that the interaction terms for sex ($p = 0.84$) or comorbidity status ($p = 0.94$) added explanatory value to the models. Similar trends were identified in the stratified models, with people with PCC demonstrating higher estimated antibody levels within the first 200 days following infection but a similar waning trajectory to those with an acute infection only. Confidence intervals for estimated levels overlapped between people with and without PCC in the subgroup who had comorbidities and in the later part of follow up across models. This reduced magnitude of difference by PCC status is likely to reflect sample size constraints - particularly during later periods of follow-up (illustrated by line chart in Fig. 3) - but may reflect between group differences for comorbidities. Between-subgroup differences in levels according to sex and comorbidity status were in line with previous research and have been investigated directly elsewhere in Virus Watch using appropriate methodology[10].

### Log Anti-S Antibody Levels over Time by PCC Status

Pre-infection anti-S responses to vaccination did not differ beyond the effects of chance in the 200 days following the second or third dose of COVID-19 vaccine (Fig. 4), illustrated by overlapping estimated levels for both groups across the full follow-up period. Levels decreased over time following both doses, with an increase towards the end of post-second dose follow-up after log quadratic decline possibly reflecting unrecorded booster vaccination. There was a log-linear decline over time following the third dose. Similarly, there were no differences beyond the effect of chance in post-infection anti-S trajectories between participants with PCC versus those with an acute infection following two-dose or three-dose vaccination in any model (Supplementary Fig. 6).

## Discussion

We aimed to investigate differences in anti-N and anti-S antibody trajectories in people who developed PCC following a mild-moderate primary SARS-CoV-2 infection versus those with an acute infection only. While participants who developed PCC had 1.8x the odds of seroconversion for anti-N and demonstrated persistently higher levels of seropositivity compared to participants with an acute infection only, stratified and adjusted analyses indicated that these differences appeared only amongst unvaccinated people and those infected with pre-Omicron variants. Anti-N antibody levels were elevated in PCC, although anti-N trajectories were similar between the two groups and

were characterised by an initial exponential increase followed by a gradual decline. Differences in anti-N antibody levels were less marked in the PCC group compared to the acute infection-only group after accounting for vaccination status and variant of infection, although differences persisted in the 200 days following infection. These may reflect a genuine difference in levels amongst seroconverters, but we cannot exclude residual confounding - in particular by infection severity which was not directly measured in this study amongst mild-moderate community cases. Notably, the current study enabled comparison of pre-infection anti-S responses to vaccination and found no evidence of a difference in levels or trajectory post-vaccination suggestive of pre-infection dysregulation in the PCC group. Similarly, post-vaccination, post-infection anti-S levels did not differ according to PCC status.

Our findings broadly corroborate smaller community-based serological studies of early pandemic infections, which found evidence of stronger antibody responses in people with PCC compared to those with acute infections only[11-13]. However, our findings indicated that this relationship was attenuated in later pandemic periods and following COVID-19 vaccination - demonstrating the importance of longitudinal studies including multiple pandemic periods in PCC research. These findings may reflect confounding by infection severity within the community cases in this study and previous literature. While all participants in this cohort experienced mild-to-moderate infections not requiring emergency care or hospitalisation, accounting for symptom severity within the mild-moderate range was not possible and is warranted. These findings may also reflect genuine differences in the mechanisms driving PCC development between early, unvaccinated cases and later vaccinated Omicron cases. This is plausible given different immunological and pathophysiological responses to infection according to variant and vaccination status[16,17] as well as differential likelihood of developing PCC amongst mild-to-moderate cases[18]. The elevated, persistent anti-N response in people with PCC - particularly amongst early, unvaccinated - cases is suggestive of a stronger response to the acute infection, which could be driven by higher viral loads and/or a pro-inflammatory state[11]. Persistent viral antigens have been suggested as another immunological mechanism elsewhere[11], though the similar waning trajectories between people with and without PCC in this group are not suggestive of persistent antigen exposure in this study. Analysing isotypes (IgA, IgG) could inform how inflammatory responses may be stimulated. Analysing IgG subclasses may be informative, as they are linked to different effector functions[19,20] and the immunoglobulin glycosylation profiles, which are associated to inflammation and metabolic health[19]. Further disaggregation of variant-specific differences investigating more granular pre-Omicron variants and accounting for symptom severity within the mild-moderate range was not possible in this study and is indicated.

Strengths of this study included the community-based sample drawn from across England, which was, to our knowledge, the largest PCC-related serological follow-up study to date. PCC was classified according to the WHO consensus definition. The cohort spanned infections beyond the first pandemic wave and enabled delineating the response to infection and vaccination as well as adjustment for demographic and clinical factors. Self-collection of serological samples enabled monthly sampling which provided more granular data around antibody trajectories than available in studies with less frequent follow-up.

**Table 2 | Predicted Probabilities of Anti-Nucleocapsid Seroconversion Stratified by Sex and Comorbidity Status (Overall Across Full Follow-Up Period)**

|  | Predicted Probability (95% CI) | |
| --- | --- | --- |
|  | **PCC** | **Acute-Only** |
| **Sex** | | |
| Female | 82.3% (75.0%, 87.9%) | 72.0% (66.0%, 77.3%) |
| Male | 90.7% (79.6%, 96.1%) | 80.2% (73.5%, 85.6%) |
| Comorbidities | | |
| Yes | 81.0% (72.1%, 87.5%) | 72.0% (64.1%, 78.8%) |
| No | 88.9% (80.6%, 93.9%) | 77.1% (71.7%, 81.8%) |
| Vaccination Status* | | |
| Unvaccinated | 88.9% (80.1%, 94.1%) | 72.0% (56.8%, 83.4%) |
| 2 Doses | 85.1% (78.9%, 89.8%) | 77.6% (72.4%, 82.0%) |
| 3 Doses | 80.4% (65.8%, 89.8%) | 82.3% (74.6%, 88.0%) |
| Variant* | | |
| Pre-Omicron | 89.6% (81.8%, 94.3%) | 74.3% (66.3%, 81.0%) |
| Omicron | 75.6% (58.8%, 87.0%) | 83.3% (74.6%, 89.4%) |

**Note:** 95% CI = 95% confidence interval; PCC = Post-Covid Condition
* Mutually adjusted for vaccination status and variant of infection.

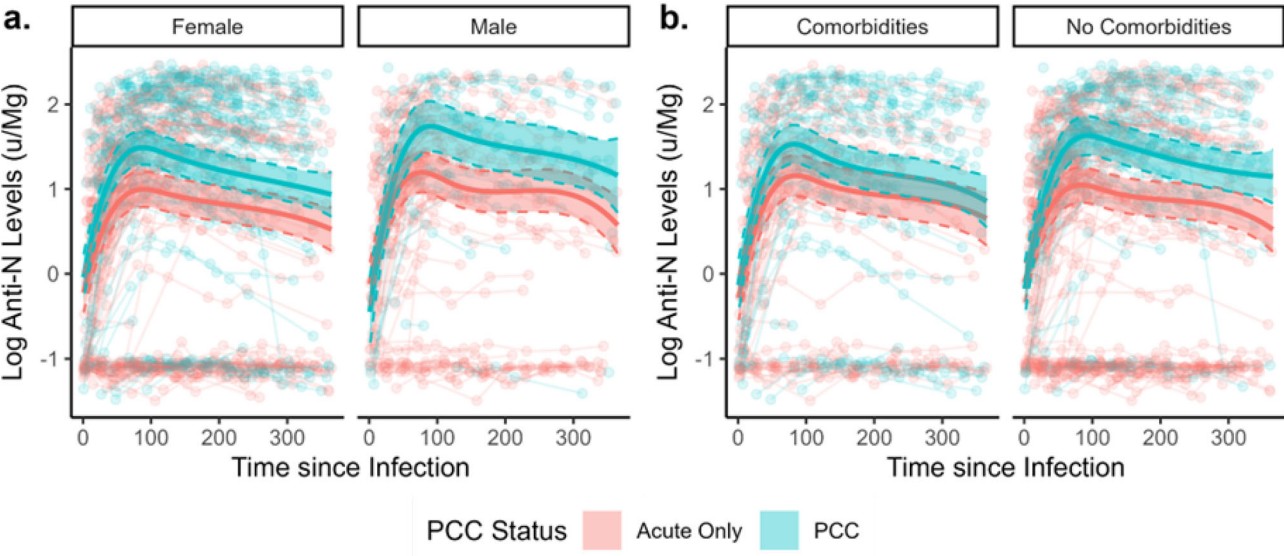

**Fig. 3 | Anti-Nucleocapsid Antibody Trajectories with 95% Confidence Intervals in Participants with Post-COVID Condition versus Acute Infection Only.** Panels present findings stratified by sex (**a**) and comorbidity status (**b**). **Note:** The legend applies to all panels in the figure.

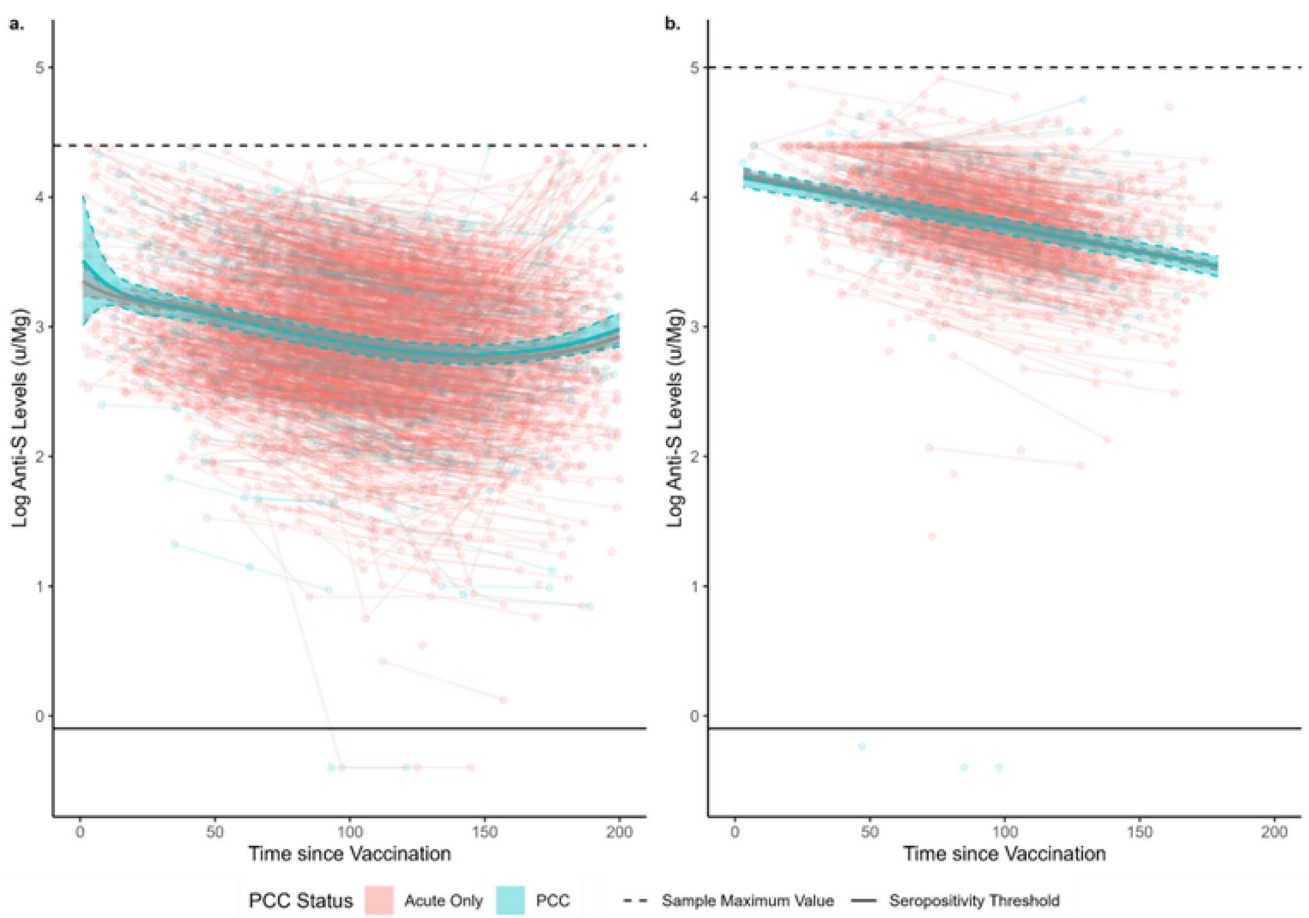

**Fig. 4 | Vaccine-Induced (Pre-Infection) Anti-Spike Antibody Trajectories with 95% Confidence Intervals in Participants who Later Developed Post-COVID Condition versus Acute Infection Only.** Panels present trajectories following two-dose (**a**) and three-dose (**b**) vaccination.

While the overall sample size of this study was large in the context of serological PCC research, subgroup sizes were relatively small over time - particularly at later follow-up time points - and it was not possible to conduct analyses stratified by more granular demographic and clinical characteristics, notably specific clinical conditions that may affect the immune response, smoking status, variant of infection, and features/phenotypes of PCC. Phenotypes of PCC were beyond the scope of this study and relevant analyses are planned within the cohort. Other key limitations included the early threshold on anti-S testing precluding comparison of anti-S antibody trajectories in unvaccinated participants and those who had received a single dose of vaccination. This study focused on primary infections. However, some infections may have been misclassified reinfections due to the lack of testing early in the pandemic; detection of later reinfections was supported by access to swab testing and detection through serology in this cohort though undetected reinfections remain possible. This study was based on mild community cases and may not generalise to more severe cases or people who received specific COVID-19 treatments. While age, sex, and comorbidities were accounted for in regressions and the average marginal effect estimates produced from these models, the sample was not representative of the English population. The sub-cohort of participants who self-selected to complete the PCC survey and who provided serological samples may be particularly motivated regarding pandemic-related health behaviours including testing and symptom monitoring, or motivated by long-term symptomatology; while this may have impacted the composition of the sample and prevalence of people with PCC, its impact on antibody trajectories is likely minimal. The follow-up period was limited to

365 days for anti-N and shorter periods for anti-S due to model stratification, and longer follow-up with monitoring of PCC symptomatology is recommended.

Based on a sample of community SARS-CoV-2 cases in England, vaccination and variant of infection appear to play an important role in driving differential likelihood of anti-N seroconversion and elevated anti-N levels amongst people with PCC. Pre-infection anti-S responses to vaccination also did not differ between people who later developed PCC following primary infection and those who recovered fully from their primary infection. These findings extend previous evidence suggesting altered adaptive immune responses in PCC, and highlight the importance of longitudinal studies including multiple pandemic and post-pandemic periods. Further investigation disaggregating the impact of infection severity at the mild-moderate level from other features of infection-related inflammation and immune activation is warranted to understand variant-related and vaccination-related differences. A nuanced understanding of underlying immune processes is crucial to developing targeted interventions and evidence-based treatments for PCC, particularly in the context of persistent disability following early infections, as well as evolving variants and widespread vaccination.

## Methods
### Ethics Approval and Consent
Virus Watch was approved by the Hampstead NHS Health Research Authority Ethics Committee: 20/HRA/2320, and conformed to the ethical standards set out in the Declaration of Helsinki. All participants provided informed consent for all aspects of the study.

## Participants

Participants (n = 2010) were a sub-cohort of Virus Watch (n = 58,628), a household longitudinal cohort study of SARS-CoV-2 infections in England and Wales running since June 2020. Recruitment and methodology of the full cohort have been described in detail elsewhere[21,22].

Recruitment criteria into the Virus Watch study were residence in England or Wales, household size up to six people with consent or assent of all household members, access to an email address, and ability to complete English-language surveys. The source population were households from the general population across all national regions of England and Wales. Households were recruited using several methods in order to achieve a target sample of > 50,000 people from the general population and representation of socioeconomically and ethnically diverse group; these included postal recruitment using a probability sample derived from the Royal Mail Post Office Address File, as well as including letter and SMS-based recruitment supported by general practices, and social media campaigns. Participants completed a detailed baseline questionnaire about demographic and clinical features for all household members, and subsequently completed weekly questionnaires about acute symptoms, SARS-CoV-2 tests, and vaccinations, and monthly questionnaires about detailed psychosocial and clinical topics tailored to the phase of the pandemic. A sub-cohort (n = 19,555) of participants over 18 years of age and resident in England also completed monthly finger-prick antibody testing for SARS-CoV-2 antibodies (see Outcomes section below) with samples collected during the period between 24/02/2021-23/03/2022; participants in the current study were drawn from this sub-cohort.

Further inclusion criteria for the current study were:

1. returned at least one finger-prick antibody sample with a valid result for anti-S and/or anti-N antibodies,
2. completed survey(s) about new-onset long-term symptoms which covered symptom development between February 2020 and March 2023, and had binary classifiable PCC status (see Exposure section below),
3. had their first recorded SARS-CoV-2 infection detected via polymerase chain reaction (PCR) or lateral flow test (LFT) before end of serological follow-up,
4. mild-moderate infection (i.e., convalesced in the community without hospitalisation).

## Exposure

We conducted a nested longitudinal case-control study, as we were interested in the difference between antibody seropositivity and trajectories over time by PCC status. The primary exposures were binary PCC status and the interaction between PCC status and time since immunogenic events (i.e., infection and - where relevant - vaccination).

**PCC Status.** Participants were classified as having developed PCC if they reported one or more new-onset long-term symptoms following polymerase chain reaction (PCR) or lateral flow test (LFT) confirmed primary infection, with symptoms meeting the World Health Organisation consensus definition for PCC: interfering long-term symptom(s) which cannot be explained by another diagnosis with an onset within three months of SARS-CoV-2 infection and a duration of at least two months[23]. Participants who had new-onset symptoms with a duration of less than two months or that developed outside of the three-month period following infection were excluded as they did not meet the WHO consensus definition of PCC but may have an immunologically distinct profile compared to those with an acute infection and no symptoms. There were insufficient numbers of these participants for detailed analysis (n = 49 not meeting duration criterion and n = 118 not meeting onset date criterion); participants not meeting onset date criterion were also excluded to prevent misclassification due to asymptomatic or undetected infection. If participants reported at least one symptom meeting the WHO consensus definition, they were classified as having PCC regardless of the duration or timing of any other reported symptoms. Participants were classified as not having developed PCC if they completed all long-term symptom surveys and never reported new-onset symptoms at any point during follow-up. Complete follow-up was required for acute-only participants to minimise misclassification of undetected new-onset symptoms.

To determine PCC status, participants were sent a questionnaire about new-onset long-term symptoms as part of the Virus Watch monthly surveys. These surveys requested participants to indicate whether they had experienced any new-onset long-term symptoms that could not be explained by another diagnosis or health state (e.g. other chronic illness or pregnancy) and provide onset dates and duration of interfering symptoms (see[18] for further detail), which were used to determine PCC status according to the criteria above. The survey did not specify that these symptoms were linked to a SARS-CoV-2 infection, to avoid perceptions of PCC influencing participants' answers. The survey was sent online to the Virus Watch cohort four times: in February 2021, May 2021, March 2022, and March 2023. Participants were asked to report new-onset long-term symptoms that developed within the previous year - covering the period between February 2020 and March 2023, which included the full period associated with antibody testing; only the May 2021 survey had a different recall period (from February 2020) as it was intended to supplement any non-response to the initial February 2021 survey regarding the previous year. While there was consequently some overlap in survey periods, symptoms could be matched by their onset date and consequently tracked if these overlapped across surveys.

**Time since immunogenic events.** Time was defined as the number of days between an immunogenic event and the antibody test date, with the event varying depending on the outcome of interest (anti-N and anti-S). The date range for sample inclusion began at 0 (i.e. day of immunogenic event) and the upper limit was determined depending on the outcome, with anti-N models capped at 365 days while the anti-S models were capped at a maximum of 200 days due to stratification (see Statistical Analysis section below). For logistic models (see Statistical Analysis section), time since immunogenic event was categorised into the following bands previously used in Virus Watch research related to seroconversion to facilitate interpretation of odds ratios by time period[10]: 0–29 days, 30–59 days, 60–89 days, 90–119 days, 120–269 days, 270+ days. In linear regression models, time in days was used to produce estimated antibody trajectories.

Infection was defined as evidence of first infection based on PCR or LFT based on linkage to UK national testing records or study-specific testing records. All participants had results available from linkage and also self-reported any SARS-CoV-2 tests taken across the study period in the weekly survey. PCR and LFT testing was also provided by the Virus Watch study during several periods, with the protocol varying over time (please see ref. 18 for details).

COVID-19 vaccination status was determined based on linkage to UK national vaccination records as well as self-reported vaccinations collected as part of the weekly survey, and was coded as (0, 1, 2, 3 doses). Samples from participants who received additional doses - which were only available to a minority of the UK population[24] - were excluded after the third dose.

Supplementary Fig. 1 illustrates data collection and classification of PCC status; serological sampling is described further below and associated analyses are illustrated in Supplementary Fig. 2.

## Outcome

The outcomes of interest were anti-N and anti-S antibody levels, based on self-collected capillary blood samples (400–600 μl) collected between 24/02/2021-23/03/2022. Participants collected samples at home using test kits produced by the company Thriva, and returned kits using prepaid priority postage. Serological testing was conducted in UK

Accreditation Service accredited-laboratories using the Roche Elecsys Anti-SARS-CoV electrochemiluminescence assays targeting total immunoglobulin (predominantly IgG, but also IgA and IgM) to the nucleocapsid (N) protein and the receptor binding domain in the S1 subunit of the spike protein[25]. Further details of the laboratory testing process for Virus Watch samples are detailed in previous Virus Watch papers[10,26].

Antibody levels were expressed as semi-quantitative numeric values in form of cut-off indices (COIs) and log-transformed to base 10. For anti-N antibodies, the manufacturer-recommended seropositivity threshold was ≥1.0, with a sensitivity of 97.2-99.5% and specificity of 99.8%[27–29]. Base-10 log transformed anti-N levels were included for samples taken between 0-365 days following PCR or LFT-confirmed primary infection. Samples that were collected following PCR- or LFT-confirmed reinfections were excluded due to the impact of reinfection on both antibody levels and unknown impact on long Covid symptomology; investigation into reinfections was beyond the scope of this analysis. Samples that were anti-N seropositive within 5 days following infection or that demonstrated a four-fold rise in levels between sequential samples taken beyond 120 days following primary infection were also excluded to remove otherwise undetected reinfections, based on established timelines of conversion and trajectories[30].

For anti-S antibodies, the manufacturer recommended seropositivity threshold was ≥ 0.8, with a sensitivity of 97.9-98.8% and a specificity of 100%[27–29]. Anti-S levels were subject to detection limits that changed over time to allow investigation into quantitative antibody levels in the highly vaccinated UK population, with limits changing from 250 u/mL between 24/02/2021 - 30/06/2021 (excluding a two-day pilot of the protocol change to increase detection limits), to 25000 u/mL between 01/07/2021 - 01/01/2022, and to 100000 u/mL between 01/01/2022 - 21/03/2022; the assay remained consistent and increased detection limits were obtained through dilution. Samples for anti-S were only included from 01/07/2021 due to a large number of samples reaching the low initial detection limit of 250 u/mL prior to this time point. The later change in the detection threshold was addressed through stratification (see Statistical Analysis section below). Samples were included if they occurred prior to primary infection (i.e., for vaccination only models) or following primary infection and prior to any confirmed reinfection. Samples following confirmed or suspected reinfection as described above were excluded for the remainder of follow-up. As with anti-N, anti-S levels were log-transformed to base 10.

## Stratification variables and covariates

The following variables based on data collected in an online demographic survey upon study registration were used to test for effect modification and stratify models and/or included as covariates in models: self-reported sex at birth (male or female), binary comorbidity status (presence of any condition on the UK NHS/government list denoting extreme clinical vulnerability or clinical vulnerability at COVID-19[31], binary variant of infection based on infection date in reference to dominant variant in participants' region of residence (pre-Omicron versus Omicron), and vaccination status at the time of infection (unvaccinated, two doses, or three doses). Please see the Statistical Analysis section for further details.

## Statistical analysis

We used binary logistic mixed models to investigate how PCC status influenced the probability of seroconversion for anti-N. A random term was included to account for individuals submitting multiple samples. Separate models were constructed to evaluate probability of ever demonstrating anti-N seropositivity across the full follow-up period, as well as models evaluating anti-N seropositivity during the following time periods, to assess between-group differences in trajectories of seropositivity: 0-29 days, 30-59 days, 60-89 days, 90-119 days, 120-269 days, 270+ days. Results were expressed as odds ratios and predicted probabilities based on average marginal effects to facilitate between-

group comparison over time. Seroconversion was investigated for anti-N only as this was the primary outcome and non-conversion is a more prominent feature of anti-N response[10]; only 3 participants in the current study did not seroconvert for anti-S.

We used linear mixed models to investigate how PCC status influenced the trajectory of log anti-N and anti-S antibody levels. The exposure was the interaction between PCC status and time since immunologic event, and the outcome was log anti-N/anti-S antibody levels. Anti-N was modelled for all participants across 365 days of follow-up, and a sensitivity analysis was conducted including only samples from participants who seroconverted for N during the study period. As anti-S antibodies respond to both vaccination status and infection status and the combination of these events may differentially affect antibody levels, models were stratified according to these characteristics. Only samples following the increase of the cap to 25000 u/ML were included, as a substantial number of samples with the early cap (250uML) reached this threshold, possibly precluding accurate estimations of levels and between-group differences. The sample period corresponded to periods of two-dose vaccination onwards in the Virus Watch study population, so anti-S models were constructed to investigate response to two-dose and three-dose vaccination as follows: pre-infection (i.e. vaccination response only), hybrid immunity with vaccination before the infection, and hybrid immunity with vaccination after the infection. A schematic diagram of these models illustrating the timing of immunogenic events is provided in Supplementary Fig. 2. All anti-S models were capped at 200 days follow-up due to data availability and UK vaccination schedules, except for the post-second-dose anti-S model (infection most recent event), which was capped at 90 days due to low sample availability beyond this point.

We tested models with time modelled as a linear term, a quadratic term, and with a B-spline with a single knot at 120 days for anti-N[10], 30 days for post-second-dose anti-S models[32], and 14 days for post-third-dose anti-S models[33]. Models were selected based on Bayesian Information Criterion values. The spline models were used for anti-N and for anti-S post-second dose (pre-infection and post-infection with vaccination as the most recent outcome); time was modelled using a linear term for the remaining anti-S models.

**Conceptual models and effect modification.** Conceptual models underlying these analyses are presented in Supplementary Fig. 3a for anti-N and Supplementary Fig. 3b and 3c for anti-S antibodies. These analyses did not aim to estimate the causal effect of PCC status on antibody responses. Rather, differences in antibody responses to infection by PCC status were investigated to provide evidence for differential immune processes and/or viral persistence, which are proposed mechanisms for PCC development, and which could not be directly measured here (denoted as node 'U' in Supplementary Fig. 3a-c). Consequently, antibody levels provide evidence as a proxy for a hypothesised process underlying PCC and are not themselves a traditional causal exposure (i.e., a relationship between antibody responses and PCC would provide evidence of an unmeasured underlying immune process, but antibodies themselves are not believed to cause PCC directly). Pre-infection demographic and clinical features are therefore proposed to influence these unmeasured mechanisms, and are consequently conceptualised as effect modifiers rather than confounders. This is because they are proposed to influence the strength and direction of the effect (i.e. effect modification) rather than inducing a potentially spurious relationship (i.e. confounding)[34] and were consequently investigated using stratification following methodological recommendations[35]. Infection severity was limited to mild-moderate community infections within this study based on inclusion criteria and study composition. The impact of variant of infection and vaccination on infection severity was accounted for using stratification and adjustment in anti-N-related models, with further description and justification provided below. The

direct effect of vaccination on anti-S antibody responses was addressed through stratification as described in the previous section.

We consequently assessed effect modification by sex at birth, comorbidity status, binary variant (pre-Omicron vs Omicron), and vaccination status at the time of infection (unvaccinated, 2 doses, or 3 doses) for anti-N seroconversion models covering the full study period. Other vaccination status categories could not be included due to insufficient anti-N samples. We included interaction terms adding sex, comorbidity status, variant, and vaccination status and evaluated evidence of the interaction providing additional explanatory power to the model using likelihood ratio tests. Given the temporal overlap between variant of infection and vaccination status and their independent relevance to infection severity, we adjusted the vaccination-stratified model for variant and vice versa. Other demographic and clinical features, such as granular clinical conditions and smoking status, could not be included as stratification variables due to limitations on sample size and/or data availability.

Predicted antibody trajectories were similarly presented stratified by sex and comorbidity based on the inclusion of an interaction term. Due to small subgroup sizes over time according to variant and vaccination status for anti-N samples, we were not able to provide stratified waning trajectories for anti-N according to variant and vaccination status. However, given the relevance of these factors both to the immune response to infection and as a potential proxy for infection severity, we conducted an adjusted analysis of the anti-N trajectory accounting for these two factors. We also adjusted the age- and sex-stratified models for variant and vaccination status.

Interaction tests were conducted for anti-N models only as anti-S models were already stratified according to vaccination status and samples were not sufficient to meaningfully assess three-way interaction for antibody trajectory across all models and timepoints. We lacked the sample size to assess effect modification for more granular variables, such as specific pre-Omicron variants of infection.

Pre-infection, vaccination-related anti-S responses were also investigated by PCC status (i.e., following later infection) to evaluate evidence for any pre-infection differences in immune response following challenge with the SARS-CoV-2 spike protein. The associated conceptual model is illustrated in Supplementary Fig. 3c. As described for the infection-related models, demographic and clinical features are appropriately conceptualised as effect modifiers within this framework.

### Reporting summary
Further information on research design is available in the Nature Portfolio Reporting Summary linked to this article.

## Data availability
Individual record serological data used in this study have been deposited in the Office of National Statistics Secure Research Service (ONS SRS) database under accession code 89201 https://ons.metadata.works/browser/dataset/89201. The Virus Watch study data are available under restricted access due to ethical and legal restrictions around individual-level health-related sensitive data, access can be obtained by following the procedure described in the ONS SRS website link about. The raw Virus Watch data are protected and are not available due to data privacy laws.

## Code availability
We have provided de-identified example code which adhere to the principles of data protection required by the project at the following link - https://doi.org/10.17605/OSF.IO/TA2BU.

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

## Acknowledgements

Virus Watch was supported by the Medical Research Council [Grant Ref: MC_PC 19070 and MR/V028375/1] awarded to ACH. The study also received $15 000 of advertising credit from Facebook to support a pilot social media recruitment campaign on 18 August 2020 awarded to ACH. The antibody testing was also supported by funding from the Department of Health and Social Care from February 2021 to March 2022 awarded to ACH. This study was also supported by the Wellcome Trust through a Wellcome Clinical Research Career Development Fellowship to R.W.A. [206602]. G.M. was supported by RYC 2020–029886-I/ AEI/ 10.13039/501100011033, co-funded by European Social Fund (ESF). From 1 May 2022, Virus Watch received funding from the European Union (Project: 101046314) awarded to IA. Views and opinions expressed are however those of the author(s) only and do not necessarily reflect those of the European Union or the European Health and Digital Executive Agency (HaDEA). Neither the European Union nor the granting authority can be held responsible for them.

## Author contributions

S.B. conceptualised the presented idea, and S.B., A.Y. and R.W.A. informed the theory. G.M. provided subject expert input regarding immunology. S.B. performed the analyses and A.Y. and R.A. verified the methods. A.Y., W.L.E.F., and V.G.N. provided data and software-related assistance. J.K. provided project management. A.C.H., I.A., and R.W.A. provided supervision. All authors contributed to the interpretation and to the editing of the final manuscript.

## Competing interests

ACH serves on the UK New and Emerging Respiratory Virus Threats Advisory Group. All other authors declare no competing interests.

## Additional information

**Sarah Beale** [1] ✉, **Alexei Yavlinsky** [1], **Gemma Moncunill** [2,3], **Wing Lam Erica Fong** [1], **Vincent Grigori Nguyen** [1,4,5], **Jana Kovar** [1], **Andrew C. Hayward** [4], **Ibrahim Abubakar** [6] & **Robert W. Aldridge** [1,7]

[1]Institute of Health Informatics, University College London, London, UK. [2]ISGlobal, Barcelona, Spain. [3]CIBER de Enfermedades Infecciosas (CIBERINFEC), Instituto de Salud Carlos III, Barcelona, Spain. [4]Institute of Epidemiology and Health Care, University College London, London, UK. [5]Department of Population, Policy and Practice, UCL Great Ormond Street Institute of Child Health, London, UK. [6]Faculty of Population Health Sciences, University College London, London, UK. [7]The Institute for Health Metrics and Evaluation, University of Washington, Seattle, USA. ✉e-mail: sarah.beale@ucl.ac.uk

