## [Peer Review file · Nature Communications]

Anti-Nucleocapsid and Anti-Spike Antibody Trajectories in People with Post-Covid Condition versus Acute-Only Infections: A Nested Longitudinal Case-Control Study within the Virus Watch Prospective Cohort

Corresponding Author: Dr Sarah Beale

Version 0:

Reviewer comments:

Reviewer #1

(Remarks to the Author)

I appreciate the opportunity to review this paper. The strengths and uniqueness of this paper include a high-quality serological follow-up data up to 365 days of COVID infection, presentation of trajectories of both anti-N and anti-S antibodies, especially pre-infection anti-S responses to vaccination. Although there are many things to appreciate with this study, I have several major concerns the extent to which those findings will be causally relevant to the Long COVID phenotype rather than underlying health conditions that may cause the same symptoms as Long COVID.

1. The author designed a case-control study nested in an existing cohort. The title is somewhat misleading because it implies this is a prospective study, but technically it is not. This is fundamental study element and should be made clear ideally both in the title and the methods section.
2. PCC in the current study was defined as the exposure, whereas anti-N and anti-S antibodies were defined as the outcomes. Does this logically make sense (PCC leading to changes in anti-N and anti-S antibodies)? Based on the introduction provided, the hypothesis should be that anti-N and anti-S antibodies reactions (exposure) lead to changes in PCC (outcome), right?
3. Inferring causality with observational data requires lots of considerations to avoid biased findings. This seems hardly achievable in the analysis with only a few covariates collected and insufficiently adjusted, especially regarding participants' comorbidities profiles (e.g. comorbidity only available as a binary variable). As such, this study should be closer to descriptive epi in my view, and the tone/wording of the conclusion should be moderated.
4. Defining Long COVID based on symptoms data poses many challenges, notably the accuracy of timing of symptom onset based on self-reported data, which is likely subject to recall bias. By only reading the texts, it is very difficult for me to conceptualize how the author utilized their data to define the PCC outcome and minimise the outcome misclassification. A visualization plot of the study design including key timelines such as infection date, antibody measurement date, survey date, and symptom onset date will be extremely helpful.
5. The authors referred to number 19 several times in the manuscript for important methodological details. Unfortunately, the hyperlink seems invalid and cannot be opened, which prevents me understanding thoroughly of the study.
6. Would you be able to provide the frequency of each condition to support the validity of the survey? If you assume that an elevated immune response is the common driver for these diverse conditions, could you articulate this in the Introduction and provide supporting evidence for it?
7. Is it possible for the authors to perform the same analysis but among people without COVID-19 infection (rather than those with COVID-19 infection)? This could act as a negative-control outcome, as we expect anti-N/S would only link to symptoms

(PCC) among the infected individuals and disappear in the uninfected. If confirmed, the evidence from this study would be much more convincing than the current.

8. The reduction from a sub-cohort of 19,555 participants, who completed monthly finger-prick antibody testing for SARS-CoV-2, to a final sample size of 2,010 seems significant. A flowchart detailing the number of exclusions for each criterion would be helpful.

9. To establish the trajectory in the mixed effects model, I assume you included only those with at least three finger-prick antibody samples yielding valid results for anti-S and/or anti-N antibodies. Does this mean the number of participants was fewer than 2,010? Could you provide the sample size used in the mixed models and describe whether this subset has specific demographic or clinical features that distinguish them from the larger cohort.

10. Did you include participants who completed all surveys about symptoms, or were participants who completed only one survey also included? If the latter, how do you account for a long-term (>2 months) symptom that began at the end of a survey period and might not be captured as PCC if only the first month is covered by that survey?

11. Why participants with new-onset symptoms lasting less than two months or developing outside the three-month period post-infection were excluded? What proportion of participants excluded due to these criteria? Why these participants were not classified as not having developed PCC?

12. If a participant had one symptom lasting less than two months and another symptom lasting more than two months, would they be excluded or remain in the study?

13. In the discussion, the authors referenced the study by Phetsouphanh et al. and hypothesized that the reduction in between-group variation in antibody titres might be due to vaccination. Your Table 1 shows that a large number of participants received three doses of vaccination. Could you stratify the analysis by vaccination status or adjust for vaccination in the model analyzing the association between anti-N and PCC to provide further insight into this hypothesis?

14. The conceptual models illustrated in Supplementary Figure 2 should include arrows from age, sex, and pre-infection health status to PCC status, since PCC is self-reported and individuals with chronic conditions may report symptoms without significant changes in immune response.

15. The severity of infection could be a major confounder, although you've addressed this by limiting the sample to mild-moderate infections. Could you further address this by restricting it to the Omicron variant, since the majority of participants were infected with it (as shown in Table 1)?

16. The trends shown in Figure 2 and Figure 3 appear inconsistent. The discrepancy in anti-N seropositivity appears to widen over time, whereas the discrepancy in anti-N levels seems to peak around day 100 post-infection and then reduces.

Reviewer #2

(Remarks to the Author)

Reviewer #3

(Remarks to the Author)

This paper addressed the question if and to what extent the humoral immune response varies in persons suffering from PCC versus those who do not. The paper is based on one of the most important infection cohorts, well written and the methods and conclusions seem appropriate. The study adds importantly to the existing body of evidence. There are number of points that could be clarified a bit more, in particular with respect to the conceptual models that focus entirely on the immunologic pathogenesis and immunologic parameters captured by the humoral response to SARS-CoV-2.

The recruiting strategy is well described but could there be a bit more information on the source population and how the households were sampled from that source population? Other papers of the Virus Watch cohort may describe this but as a service to readers it would be nice to briefly describe it here as well.

The conceptual models presented in the supplement are helpful and authors and journals should be encouraged to follow this example since they are a critical basis for the statistical models. I just have a few questions: Smoking is not represented although it has a substantial effect on the antibody response (odds of seroconversion and antibody levels are lower). Is this information not available in the data sets and cannot be accounted for in the analyses? Why is infection itself not represented in conceptual models? or is this part of the node "infection severity"? Finally, why is there no direct arrow from infection severity to PCC? There are multiple pathogenetic paths, not just through the immunological path triggered by the infection, and infection severity is associated with PCC (though maybe not strongly in this data set since restricted to mild to moderate infections). This latter point is particularly important since it refers to the point made above that the analyses focus entirely on the immunologic pathogenesis and immunologic parameters captured by the humoral response to SARS-CoV-2. The analyses seem appropriate to address the study questions and to deal with complex nature of the data. However, there are several ways to plan and conduct the analyses and the assumptions for them. To increase comparability with other

studies and also for serving as excellent example for the analyses to address the question of how immunologic trajectories vary in persons with and without PCC it would be useful to be even more detailed about the assumptions/decisions taken (and why preferred over others) and to share the codes. The literature is and will be heterogeneous for this question and some alignment of the analyses may help to reduce heterogeneity but also better understand why results differ. There is one sentence in the conclusion section that is not really backed up by the data of this study ("These findings build upon previous evidence around altered adaptive immune responses in PCC 620 suggestive of chronic post-infection inflammation and immune activation and/or viral persistence in people with PCC"). Neither immune activation nor viral persistence was assessed in this study. It is appropriate to refer to that in the discussion section but it seems a bit of a long shot in the conclusion section.

Reviewer #4

(Remarks to the Author)

Anti-Nucleocapsid and Anti-Spike Antibody Trajectories in People with Post-Covid Condition versus Acute-Only Infections: Results from the Virus Watch Prospective Cohort Study
by Sarah Beale et al

This study use data from the large "Virus Watch" study that was a longitudinal study of household SARS-CoV-2 infections from 2020 and onwards.

This paper utilize that a subset of participants had monthly finger-prick blood draw performed to evaluate SARS-CoV-2 antibody development. In this sub-group the authors identify individuals that had persistent symptoms meeting the criteria (WHO) for classification as long-covid / post covid condition (PCC).

The main question for the paper was then to evaluate seroconversion and anti-nucleocapsid trajectories in the PCC group vs acute only group.

The main finding reported is that individuals classified as PCC have higher rates of nucleocapsid seroconversion and higher levels of anti-nucleocapsid antibodies following infection. This is in contrast to anti-Spike antibodies induced via vaccination.

The paper is well written and methodological sound.

However, I have one major concern. Table 1 display the participant characteristics. My main concern here is the large differences in the Acute only group vs the post covid condition group in that 25% of the PCC participants are unvaccinated whereas only 3.7% in the acute only group are non-vaccinated.

COVID-19 vaccination has repeatedly been shown to both reduce the likelihood of seroconversion on nucleocapsid and in confirmed infections (with seroconversion) to mute the nucleocapsid response. It looks like the authors are aware of this (as shown in conceptual analysis supplementary figure 2A.) but I fail to see how this impacted the results. Sex, age and comorbidities were assessed as effect modifier. Why only these demographics?

Thus, this important component and skewing in the two groups need to be investigated to confirm or refute the results. I am aware that significant power will be lost but vaccination is really an important modifier of nucleocapsid antibody levels.

The authors perform an analysis evaluating only those that seroconverted in supplementary figure 4 and some differences persist but levels in the two groups converge after 200 days. The impact of vaccination is not discussed in the text despite being different in the two groups and shown in the conceptual analysis plot to be important

The fact that the trajectories of decline in nucleocapsid antibody levels are similar in the acute only vs PPC group does not suggest that there are any differences in antigen exposure or inflammation/activation beyond the acute infection period.

The strength of the setup is the ability in this rather large longitudinal cohort to compare immune responses to vaccination and response to infection (presumed inducer of PCC-status) and here the authors show no differences in anti-spike responses arguing for no different immune-competence between the two groups.

Minor. The initial description of the collection of survey results and the antibody testing period is hard to follow. Example line 229-232: Participants were asked to report new-onset long-term symptoms that developed within the previous year – covering the period between February 2020 and March 2023, which included the full period associated with antibody testing;

Further up in the description of antibody testing was performed from early 2021 onwards...

Version 1:

Reviewer comments:

Reviewer #1

(Remarks to the Author)

Thank you to the authors for their work on revising the manuscript. But, I am afraid I am still concerned with the current interpretation of key message from this study.

The main issue is the causal framework used to analyse the relationship between antibody response and PCC development. The authors acknowledged the limitations in establishing causality (as noted in their response to Comment 3), they continue to use causal approaches like "Conceptual Models".

The Conceptual Models contain several questionable assumptions. For example, in Supplementary Figure 3, these key variables like age, sex, smoking status, and pre-infection health are all treated as effect modifiers rather than confounders. This mischaracterization is very misleading, particularly for non-technique readers. Even if these factors could partially moderate the pathway, the analysis should control for them and provide adjusted estimates.

Of note, the baseline characteristics in Table 1 shows important differences between groups. PCC patients were obviously younger and more female. These factors are well known to correlate with stronger immune responses, which could explain a lot why the higher and persistent anti-N antibody levels observed in PCC patients. Without accounting for these potential confounders, it is difficult to draw meaningful mechanistic insights about Long COVID from the current study but making more confusion/noise.

Reviewer #3

(Remarks to the Author)

There were a lot of questions which addressed many aspects of the study. I think the authors answered these questions well and revised the paper accordingly. Also, I appreciate the more cautious wording and for putting the results of this study well in the context of the existing evidence. I see no further need for revisions.

Reviewer #4

(Remarks to the Author)

Evaluation of a revised manuscript: "Anti-Nucleocapsid and Anti-Spike Antibody Trajectories in People with Post-Covid Condition versus Acute-Only Infections: Results from a Nested Longitudinal Case-Control Study within the Virus Watch Prospective Cohort" by Beale et al.

The authors have very comprehensively discussed the points raised and prepared additional analysis to corroborate their results.

Specifically I find the adjustment of vaccination status very important in this context as it alludes to an immune driven mechanism of PCC.

I think the additional analysis and manuscript additions are sufficient

Response to Reviewer's Comments

Thank you for the opportunity to submit a revision of our manuscript regarding anti-Nucleocapsid and anti-Spike antibody responses and Post-Covid Condition. Based on the reviewers' thoughtful comments, we have substantially revised the manuscript, including adding additional analyses accounting for the impact of variant and vaccination status, providing further methodological information and clarification in the Methods section, and revising the discussion section given the additional analyses and in light of other comments. We have provided a point-by-point response to the reviewers' comments below, with our responses provided in italicised text.

Reviewer #1 (Remarks to the Author):

I appreciate the opportunity to review this paper. The strengths and uniqueness of this paper include a high-quality serological follow-up data up to 365 days of COVID infection, presentation of trajectories of both anti-N and anti-S antibodies, especially pre-infection anti-S responses to vaccination. Although there are many things to appreciate with this study, I have several major concerns the extent to which those findings will be causally relevant to the Long COVID phenotype rather than underlying health conditions that may cause the same symptoms as Long COVID.

1. The author designed a case-control study nested in an existing cohort. The title is somewhat misleading because it implies this is a prospective study, but technically it is not. This is fundamental study element and should be made clear ideally both in the title and the methods section.

We have amended the title as follows to make this study element clearer, as well as highlight the longitudinal nature of the data underlying the current study compared to traditional case-control designs: "Anti-Nucleocapsid and Anti-Spike Antibody Trajectories in People with Post-Covid Condition versus Acute-Only Infections: Results from a Nested Longitudinal Case-Control Study within the Virus Watch Prospective Cohort"

We added the following clarification to Methods section (paragraph 5): "We conducted a nested longitudinal case-control study, as we were interested in the difference between antibody levels and trajectories over time by PCC status. ..."

2. PCC in the current study was defined as the exposure, whereas anti-N and anti-S antibodies were defined as the outcomes. Does this logically make sense (PCC leading to changes in anti-N and anti-S antibodies)? Based on the introduction provided, the hypothesis should be that anti-N and anti-S antibodies reactions (exposure) lead to changes in PCC (outcome), right?

Our study is guided by a conceptual model based on preliminary evidence that an aberrant immune response may underlie both PCC and differences in antibody trajectories, and that antibody trajectories are consequently a useful indicator to test this hypothesis, but not themselves a cause. This is introduced within the first paragraph of the Introduction and then elaborated in the following paragraphs, with explicit assumptions within the current study further described in the 'Conceptual Model...' section of the Methods. By defining PCC as the exposure and antibody levels as the outcomes, we aim to explore the association between PCC status and antibody dynamics over time, as specified in the Aims and Objectives. Further detail around the conceptual approach including justification is then provided in the 'Conceptual Models and Effect Modification' section of the Methods.

3. Inferring causality with observational data requires lots of considerations to avoid biased findings. This seems hardly achievable in the analysis with only a few covariates collected and insufficiently adjusted, especially regarding participants' comorbidities profiles (e.g. comorbidity only available as a binary variable). As such, this study should be closer to descriptive epi in my view, and the tone/wording of the conclusion should be moderated.

We agree with the reviewers that this study is closer to descriptive epidemiology and does not aim to infer causality, but rather to build an evidence case around immune responses in PCC and acute-only Covid-19. We state this explicitly in the 'Conceptual Models and Effect Modification' section of the methods: "These analyses did not aim to estimate the causal effect of PCC status on antibody responses. Rather, differences in antibody responses to infection by PCC status were investigated to provide evidence for differential immune processes and/or viral persistence, which are proposed mechanisms for PCC development and which could not be directly measured here (denoted as node 'U' in Supplementary Figure 2a-c)..."

We have revised the wording of the Conclusion section to ensure that results are not presented using causal language.

4. Defining Long COVID based on symptoms data poses many challenges, notably the accuracy of timing of symptom onset based on self-reported data, which is likely subject to recall bias. By only reading the texts, it is very difficult for me to conceptualize how the author utilized their data to define the PCC outcome and minimise the outcome misclassification. A visualization plot of the study design including key timelines such as infection date, antibody measurement date, survey date, and symptom onset date will be extremely helpful.

Thank you for this suggestion – we have now added a figure (Supplementary Figure 1) which is referenced in the 'Exposure' section of the Methods, in order to visualise the study design and its relationship to defining Post-Covid Condition status. Antibody measurement dates are included in this figure, with further illustration of serology within the context of the models illustrated in Supplementary Figure 2.

5. The authors referred to number 19 several times in the manuscript for important methodological details. Unfortunately, the hyperlink seems invalid and cannot be opened, which prevents me understanding thoroughly of the study.

We have updated the citation for number 19 in the reference list.

6. Would you be able to provide the frequency of each condition to support the validity of the survey? If you assume that an elevated immune response is the common driver for these diverse conditions, could you articulate this in the Introduction and provide supporting evidence for it?

The syndromic definition of Long Covid is provided in detail in the Exposure section of the Methods and is the only condition under investigation in this study. Frequencies are listed in Paragraph 1 of the results and Table 1. The hypothesis around immune response and persistent symptoms is articulated in Paragraphs 3 and 4 of the Introduction, including supporting evidence.

7. Is it possible for the authors to perform the same analysis but among people without COVID-19 infection (rather than those with COVID-19 infection)? This could act as a negative-control outcome, as we expect anti-N/S would only link to symptoms (PCC) among

the infected individuals and disappear in the uninfected. If confirmed, the evidence from this study would be much more convincing than the current.

We appreciate the reviewer's suggestion. However, we believe this is not methodologically appropriate in this context.

Anti-N antibodies are a specific immunological marker of SARS-CoV-2 infection. By definition, individuals without COVID-19 infection would not have these antibodies. Therefore, conducting the same analysis in uninfected individuals would yield null results and would not serve as a meaningful negative-control outcome but rather demonstrate a lack of relevance for these markers in uninfected populations, which is already well-established. The expected absence of a relationship between anti-N/S antibodies and symptoms in uninfected individuals is not informative regarding the validity of our findings among the infected cohort.

Anti-S antibodies are affected by vaccination as well as infection and thus require different assumptions and models. We have already performed stratified analyses on uninfected, vaccinated participants to account for this. These are reported in Figure 5 and Supplementary Figure 5 and in the associated text in the Results section.

8. The reduction from a sub-cohort of 19,555 participants, who completed monthly finger-prick antibody testing for SARS-CoV-2, to a final sample size of 2,010 seems significant. A flowchart detailing the number of exclusions for each criterion would be helpful.

This flowchart is given in the Supplementary Material (Supplementary Figure 3), and referenced in paragraph 1 of the Results section.

9. To establish the trajectory in the mixed effects model, I assume you included only those with at least three finger-prick antibody samples yielding valid results for anti-S and/or anti-N antibodies. Does this mean the number of participants was fewer than 2,010? Could you provide the sample size used in the mixed models and describe whether this subset has specific demographic or clinical features that distinguish them from the larger cohort.

Mixed effect models do not require three or more observations per group, so this inclusion criterion was not used, and all available data were entered as described in the Statistical Analysis section and detailed in the Supplementary Material (Supplementary Table 1). Mixed models are robust to uneven group sizes and provide estimates for trends over time that can include singleton clusters and multiple-observation clusters. Variance is partitioned into within- and between-cluster variance, with singleton clusters having no within-cluster variance but still contributing to slope estimation and estimation of total variance.

10. Did you include participants who completed all surveys about symptoms, or were participants who completed only one survey also included? If the latter, how do you account for a long-term (>2 months) symptom that began at the end of a survey period and might not be captured as PCC if only the first month is covered by that survey?

As described in the Exposure section of the Methods (paragraph 2, 'PCC Status') participants were follow-up regarding PCC status until PCC development according to the WHO consensus definition (and thus could not fall within the latter suggested case) or completion of all surveys (i.e. no gap in time period covered).

11. Why participants with new-onset symptoms lasting less than two months or developing outside the three-month period post-infection were excluded? What proportion of participants

excluded due to these criteria? Why these participants were not classified as not having developed PCC?

We have provided the following clarification in the Exposure section of the Methods ('PCC Status, paragraph 1): "Participants who had new-onset symptoms with a duration of less than two months or that developed outside of the three month period following infection were excluded as they did not meet the WHO consensus definition of PCC but may have an immunologically distinct profile compared to those with an acute infection and no symptoms. There were insufficient numbers of these participants for detailed analysis (n=49 not meeting duration criterion and n=118 not meeting onset date criterion); participants not meeting onset date criterion were also excluded to prevent misclassification due to asymptomatic or undetected infection."

12. If a participant had one symptom lasting less than two months and another symptom lasting more than two months, would they be excluded or remain in the study?

Participants with at least one symptom meeting the stipulated definition were coded as having PCC. We have added the following clarification to the 'PCC Status' subsection of the Methods: "If participants reported at least one symptom meeting the WHO consensus definition, they were classified as having PCC regardless of the duration or timing of any other reported symptoms."

13. In the discussion, the authors referenced the study by Phetsouphanh et al. and hypothesized that the reduction in between-group variation in antibody levels might be due to vaccination. Your Table 1 shows that a large number of participants received three doses of vaccination. Could you stratify the analysis by vaccination status or adjust for vaccination in the model analyzing the association between anti-N and PCC to provide further insight into this hypothesis?

We have now accounted for vaccination status in the anti-N models, with associated analytical methods described in the Methods section ('Conceptual Models and Effect Modification, para 1-3). Results are presented in Table 2 and Figures 3 and 4 and described in the associated text. Overall, accounting for vaccination status suggests that differences in seropositivity (i.e. binary detection threshold) were most prominent in unvaccinated people. However, differences in levels (i.e. anti-N levels/trajectory) persisted after accounting for vaccination. These findings are discussed in detail in paragraphs 1-2 of the Discussion, and we highlight the importance of these factors within the Conclusions section.

14. The conceptual models illustrated in Supplementary Figure 2 should include arrows from age, sex, and pre-infection health status to PCC status, since PCC is self-reported and individuals with chronic conditions may report symptoms without significant changes in immune response.

We have now provided additional detail in paragraph 2 of the questionnaire in the Exposure ('PCC Status') section of the Methods to clarify this point. The questionnaire specifically addressed new onset long-term symptoms that cannot be explained by another diagnosis or health state (e.g. chronic illness or pregnancy) – following the WHO consensus definition as stated in paragraph 1 of this section – and consequently symptoms related to chronic conditions were not eligible for report. Any new symptoms following infection met the syndromic definition of PCC as specified by the WHO. The conceptual model is also intended to describe visually the rationale of the analysis and is not intended as a Directed Acyclic Graph, which is not appropriate in this context given that the statistical analysis is not

intended to provide a causal estimate of the impact of the exposure on the outcome but rather to triangulate the broader hypothesis around adaptive immune responses.

15. The severity of infection could be a major confounder, although you've addressed this by limiting the sample to mild-moderate infections. Could you further address this by restricting it to the Omicron variant, since the majority of participants were infected with it (as shown in Table 1)?

We agree with the reviewer that stratification by variant where possible is relevant, although it may reflect variant-related differences rather than severity within this cohort of mild-moderate infections and this cannot be directly inferred from the data. We have added stratification according to binary variant status (pre-Omicron vs Omicron) as suggested for N-seroconversion as there were insufficient samples over time in the Omicron period for stratified antibody trajectory analyses (as explained in paragraph 1 of the results, “the majority of samples (79%, n= 7512) occurred prior to infection and were included in pre-infection, vaccination-related anti-S models only”). For the Anti-N level trajectory analyses, we have consequently instead adjusted models to account for variant. Analytical methods and justification are described in the Methods section in further detail (“Conceptual Models and Effect Modification, para 1-3).

Results are reported in the Results in Table 2, Supplementary Table 3, and Figures 3 and 4 and in associated text. We found evidence of an interaction between PCC status and variant (Supplementary Table 3), with people with PCC demonstrating greater odds of seroconversion compared to those with an acute infection only during the pre-Omicron period only (Table 2). For anti-N trajectories (i.e. antibody levels as opposed to passing the binary conversion threshold), we found that differences persisted after adjustment.

We have provided substantial discussion of these findings in the Discussion, including the following regarding severity in paragraph 2: “These findings may reflect confounding by infection severity within the community cases in this study and previous literature. While all participants in this cohort experienced mild-to-moderate infections not requiring emergency care or hospitalisation, accounting for symptom severity within the mild-moderate range was not possible and is warranted. These findings may also reflect genuine differences in the mechanisms driving PCC development between early, unvaccinated cases and later vaccinated Omicron cases. This is plausible given different immunological and pathophysiological responses to infection according to variant and vaccination status (33, 34) as well as differential likelihood of developing PCC amongst mild-to-moderate cases (19)...”

16. The trends shown in Figure 2 and Figure 3 appear inconsistent. The discrepancy in anti-N seropositivity appears to widen over time, whereas the discrepancy in anti-N levels seems to peak around day 100 post-infection and then reduces.

Seropositivity and antibody levels are different outcomes which are not directly comparable, which is why they were modelled separately within these analyses. Seropositivity refers to reaching a (relatively low) threshold of detectable antibody positivity, and participants can remain seropositive (i.e., detectable levels of antibody in the sera) for a long time while their levels of antibody drop. These drops can be substantial even while remaining seropositive. These results indicate that more participants in the PCC group had detectable levels of antibody for longer (i.e. seropositive), and that the levels of antibody peaked around day 100 and waned in both groups (i.e. trends in levels).

Reviewer #2 (Remarks to the Author):

Thank you. Please refer to the response to Reviewer 1 above.

Reviewer #3 (Remarks to the Author):

This paper addressed the question if and to what extent the humoral immune response varies in persons suffering from PCC versus those who do not. The paper is based on one of the most important infection cohorts, well written and the methods and conclusions seem appropriate. The study adds importantly to the existing body of evidence. There are number of points that could be clarified a bit more, in particular with respect to the conceptual models that focus entirely on the immunologic pathogenesis and immunologic parameters captured by the humoral response to SARS-CoV-2.

The recruiting strategy is well described but could there be a bit more information on the source population and how the households were sampled from that source population? Other papers of the Virus Watch cohort may describe this but as a service to readers it would be nice to briefly describe it here as well.

We have added the following brief description to the 'Participants' section of the Methods, with further description provided in the protocol and cohort profile papers cited in that section: "The source population were households from the general population across all national regions of England and Wales. Households were recruited using several methods in order to achieve a target sample of <50,000 people from the general population and representation of socioeconomically and ethnically diverse group; these included postal recruitment using a probability sample derived from the Royal Mail Post Office Address File, as well as including letter and SMS-based recruitment supported by general practices, and social media campaigns."

The conceptual models presented in the supplement are helpful and authors and journals should be encouraged to follow this example since they are a critical basis for the statistical models. I just have a few questions: Smoking is not represented although it has a substantial effect on the antibody response (odds of seroconversion and antibody levels are lower). Is this information not available in the data sets and cannot be accounted for in the analyses? Why is infection itself not represented in conceptual models? or is this part of the node "infection severity"? Finally, why is there no direct arrow from infection severity to PCC? There are multiple pathogenetic paths, not just through the immunological path triggered by the infection, and infection severity is associated with PCC (though maybe not strongly in this data set since restricted to mild to moderate infections). This latter point is particularly important since it refers to the point made above that the analyses focus entirely on the immunologic pathogenesis and immunologic parameters captured by the humoral response to SARS-CoV-2.

Thank you for raising these constructive points about the conceptual models. We have addressed all points raised in order:

- 1) *We have now included nodes within the conceptual models to address the role of smoking. We lacked sufficient data to include this variable in the analyses, and consequently have acknowledged in limitation in both the Methods section and the Limitations section as follows:*

Methods ('Conceptual Models and Effect Modification, par. 2): "Other demographic and clinical features, such as granular clinical conditions and smoking status, could not be included as stratification variables due to limitations on sample size and/or data availability."

Limitations section of Discussion (par. 2): "... it was not possible to conduct analyses stratified by more granular demographic and clinical characteristics, notably specific clinical conditions that may affect the immune response, smoking status, variant of infection, and features/phenotypes of PCC."

- 2) *Infection was not represented in the conceptual models because infection was a requirement for inclusion in the models (i.e. is an assumed prior for the model's relevance). We have included 'post-infection' within the titles of the models where relevant to clarify this point.*
- 3) *We have not included direct arrows into the PCC variables from infection severity as the conceptual model is intended to illustrate the proposed unmeasured mechanisms and their relevance to antibody responses and PCC to put the analytical choices in context. However, we agree that other pathogenic mechanisms are relevant and should be illustrated in the model. Consequently, we have now added another unmeasured node indicating 'other pathogenic mechanisms' to the post-infection models to provide clarification.*

The analyses seem appropriate to address the study questions and to deal with complex nature of the data. However, there are several ways to plan and conduct the analyses and the assumptions for them. To increase comparability with other studies and also for serving as excellent example for the analyses to address the question of how immunologic trajectories vary in persons with and without PCC it would be useful to be even more detailed about the assumptions/decisions taken (and why preferred over others) and to share the codes. The literature is and will be heterogeneous for this question and some alignment of the analyses may help to reduce heterogeneity but also better understand why results differ.

We agree with the reviewer that clear description and justification of the analytical choices is important. Consequently, have provided extensive detail about the analytical choices and assumptions in the Statistical Methods and Conceptual Models and Effect Modification sections of the Methods section, including amendments and requested clarification based on points raised by the reviewers here. We also provide more supplementary background information regarding the data and analyses in Supplementary Figures 1-3 and have now included a link to de-identified example code (see 'Code Availability').

There is one sentence in the conclusion section that is not really backed up by the data of this study ("These findings build upon previous evidence around altered adaptive immune responses in PCC 620 suggestive of chronic post-infection inflammation and immune activation and/or viral persistence in people with PCC"). Neither immune activation nor viral persistence was assessed in this study. It is appropriate to refer to that in the discussion section but it seems a bit of a long shot in the conclusion section.

We have removed this sentence in the Conclusions section and revised the section as follows: “These findings extend previous evidence suggesting altered adaptive immune responses in PCC, and highlight the importance of longitudinal studies including multiple pandemic and post-pandemic periods. Further investigation disaggregating the impact of infection severity at the mild-moderate level from other features of infection-related inflammation and immune activation is warranted to understand variant-related and vaccination-related differences...”

Reviewer #4 (Remarks to the Author):

Anti-Nucleocapsid and Anti-Spike Antibody Trajectories in People with Post-Covid Condition versus Acute-Only Infections: Results from the Virus Watch Prospective Cohort Study by Sarah Beale et al

This study use data from the large “Virus Watch” study that was a longitudinal study of household SARS-CoV-2 infections from 2020 and onwards.

This paper utilize that a subset of participants had monthly finger-prick blood draw performed to evaluate SARS-CoV-2 antibody development. In this sub-group the authors identify individuals that had persistent symptoms meeting the criteria (WHO) for classification as long-covid / post covid condition (PCC).

The main question for the paper was then to evaluate seroconversion and anti-nucleocapsid trajectories in the PCC group vs acute only group.

The main finding reported is that individuals classified as PCC have higher rates of nucleocapsid seroconversion and higher levels of anti-nucleocapsid antibodies following infection. This is in contrast to anti-Spike antibodies induced via vaccination.

The paper is well written and methodological sound.

However, I have one major concern. Table 1 display the participant characteristics. My main concern here is the large differences in the Acute only group vs the post covid condition group in that 25% of the PCC participants are unvaccinated whereas only 3.7% in the acute only group are non-vaccinated.

COVID-19 vaccination has repeatedly been shown to both reduce the likelihood of seroconversion on nucleocapsid and in confirmed infections (with seroconversion) to mute the nucleocapsid response. It looks like the authors are aware of this (as shown in conceptual analysis supplementary figure 2A.) but I fail to see how this impacted the results. Sex, age and comorbidities were assessed as effect modifier. Why only these demographics?

Thus, this important component and skewing in the two groups need to be investigated to confirm or refute the results. I am aware that significant power will be lost but vaccination is really an important modifier of nucleocapsid antibody levels.

The authors perform an analysis evaluating only those that seroconverted in supplementary figure 4 and some differences persist but levels in the two groups converge after 200 days. The impact of vaccination is not discussed in the text despite being different in the two groups and shown in the conceptual analysis plot to be important

Thank you for raising these points – we have now made significant revisions to the analyses and discussion in order to account for vaccination status (as well as variant of infection) in the anti-N models. As described in the response to Reviewer 1, relevant analytical methods are described in the Methods section (“Conceptual Models and Effect Modification, para 1-3),

and Results presented in Table 2 and Figures 3 and 4 and described in the associated text. Overall, accounting for vaccination status and variant of infection suggests that differences in seropositivity were most prominent in unvaccinated, pre-Omicron infections. However, differences in levels (i.e. anti-N levels) persisted after accounting for these factors. These findings are discussed in detail in paragraphs 1-2 of the Discussion, and we highlight the importance of these factors within the Conclusions section.

The fact that the trajectories of decline in nucleocapsid antibody levels are similar in the acute only vs PPC group does not suggest that there are any differences in antigen exposure or inflammation/activation beyond the acute infection period.

Thank you for raising this point. We have amended the Discussion, and have added the following specific clarification regarding this point in Discussion paragraph 2: "Persistent viral antigens have been suggested as another immunological mechanism elsewhere prompting the continued production of these antibodies (11), though the similar waning trajectories between people with and without PCC in this group are not suggestive of persistent antigen exposure in this study."

The strength of the setup is the ability in this rather large longitudinal cohort to compare immune responses to vaccination and response to infection (presumed inducer of PCC-status) and here the authors show no differences in anti-spike responses arguing for no different immune-competence between the two groups.

Minor. The initial description of the collection of survey results and the antibody testing period is hard to follow. Example line 229-232: Participants were asked to report new-onset long-term symptoms that developed within the previous year – covering the period between February 2020 and March 2023, which included the full period associated with antibody testing;

Further up in the description of antibody testing was performed from early 2021 onwards...

We have now added a figure (Supplementary Figure 1) - referenced in the 'Exposure' section of the methods – which illustrates data collection dates and the relationship between data sources and Post-Covid Condition Status. We believe this adds further clarity around these points. Survey dates regarding new-onset long-term conditions spanned a longer time period than serological data collection, but included the full time period during which serological data were collected (i.e. complete coverage of study period regarding new-onset long-term symptoms). Data regarding infections and new-onset symptoms (i.e., PCC Status classification) prior to the serological data collection period are relevant given that serological status reflect prior infection or vaccination, with time since immunological event addressed within the statistical modelling strategy.

Response to Reviewer's Comments

Thank you for the opportunity to provide final revisions of our manuscript regarding anti-Nucleocapsid and anti-Spike antibody responses and Post-Covid Condition. We have addressed the editorial requests, including completing the editorial checklist and providing a response to Reviewer 1's suggestions. Our responses are provided in italicised text below. The remaining reviewers did not suggest any further changes, and we thank them for their positive feedback.

Editor:

To address remaining concerns of reviewer #1, please provide a more detailed justification, including literature, for the use of conceptual models and effect modifiers (rather than confounders) (Methods line 338-340: "Consequently, pre-infection demographic...").

Reviewer #1 (Remarks to the Author):

Thank you to the authors for their work on revising the manuscript. But, I am afraid I am still concerned with the current interpretation of key message from this study.

The main issue is the causal framework used to analyse the relationship between antibody response and PCC development. The authors acknowledged the limitations in establishing causality (as noted in their response to Comment 3), they continue to use causal approaches like "Conceptual Models".

The Conceptual Models contain several questionable assumptions. For example, in Supplementary Figure 3, these key variables like age, sex, smoking status, and pre-infection health are all treated as effect modifiers rather than confounders. This mischaracterization is very misleading, particularly for non-technique readers. Even if these factors could partially moderate the pathway, the analysis should control for them and provide adjusted estimates.

Of note, the baseline characteristics in Table 1 shows important differences between groups. PCC patients were obviously younger and more female. These factors are well known to correlate with stronger immune responses, which could explain a lot why the higher and persistent anti-N antibody levels observed in PCC patients. Without accounting for these potential confounders, it is difficult to draw meaningful mechanistic insights about Long COVID from the current study but making more confusion/noise.

As requested by the editor, we provided a more detailed explanation and justification for the approach within the Methods section, including citations. We believe that the reviewer has fundamentally misunderstood the approach, which does not provide a causal exposure-outcome relationship in the traditional sense but rather uses an intermediate proxy variable to investigate evidence for the central hypothesis. The hypothesised mechanism under investigation may vary across subgroups (i.e. effect modification). We have consequently made the following clarification within the Methods to address this:

"Conceptual models underlying these analyses are presented in Supplementary Figure 3a for anti-N and Supplementary Figure 3b and 3c for anti-S antibodies. These analyses did not aim to estimate the causal effect of PCC status on antibody responses. Rather, differences in antibody responses to infection by PCC status were investigated to provide evidence for differential immune processes and/or viral persistence, which are proposed mechanisms for PCC development, and which could not be directly measured here (denoted as node 'U' in Supplementary Figure

3a-c). Consequently, antibody levels provide evidence as a proxy for a hypothesised process underlying PCC and are not themselves a traditional causal exposure (i.e., a relationship between antibody responses and PCC would provide evidence of an unmeasured underlying immune process, but antibodies themselves are not believed to cause PCC directly). Pre-infection demographic and clinical features are therefore proposed to influence these unmeasured mechanisms, and are consequently conceptualised as effect modifiers rather than confounders. This is because they are proposed to influence the strength and direction of the effect (i.e. effect modification) rather than inducing a potentially spurious relationship (i.e. confounding) (34), and were consequently investigated using stratification following methodological recommendations (35)..."